# Variational Diffusion Autoencoders with Random Walk Sampling

## Abstract

Variational inference (VI) methods and especially variational autoencoders (VAEs) specify scalable generative models that enjoy an intuitive connection to manifold learning — with many default priors the posterior/likelihood pair $q(z|x)/p(x|z)$ can be viewed as an approximate homeomorphism (and its inverse) between the data manifold and a latent Euclidean space. However, these approximations are well-documented to become degenerate in training. Unless the subjective prior is carefully chosen, the topologies of the prior and data distributions often will not match. Conversely, diffusion maps (DM) automatically *infer* the data topology and enjoy a rigorous connection to manifold learning, but do not scale easily or provide the inverse homeomorphism. In this paper, we propose **a)** a principled measure for recognizing the mismatch between data and latent distributions and **b)** a method that combines the advantages of variational inference and diffusion maps to learn a homeomorphic generative model. The measure, the *locally bi-Lipschitz property*, is a sufficient condition for a homeomorphism and easy to compute and interpret. The method, the *variational diffusion autoencoder* (VDAE), is a novel generative algorithm that first infers the topology of the data distribution, then models a diffusion random walk over the data. To achieve efficient computation in VDAEs, we use stochastic versions of both variational inference and manifold learning optimization. We prove approximation theoretic results for the dimension dependence of VDAEs, and that locally isotropic sampling in the latent space results in a random walk over the reconstructed manifold. Finally, we demonstrate our method on various real and synthetic datasets, and show that it exhibits performance superior to other generative models.

## 1 Introduction

Recent developments in generative models such as variational auto-encoders (VAEs, Kingma & Welling (2013)) and generative adversarial networks (GANs, Goodfellow et al. (2014)) have made it possible to sample remarkably realistic points from complex high dimensional distributions at low computational cost. While their methods are very different — one is derived from variational inference and the other from game theory — their ends both involve learning smooth mappings from a user-defined prior distribution to the modeled distribution.

These maps are closely tied to manifold learning when the prior is supported over a Euclidean space (e.g. Gaussian or uniform priors) and the data lie on a manifold (also known as the Manifold Hypothesis, see Narayanan & Mitter (2010); Fefferman et al. (2016)). This is because manifolds themselves are defined by sets that have homeomorphisms to such spaces. Learning such maps is beneficial to any machine learning task, and may shed light on the success of VAEs and GANs in modeling complex distributions.

Furthermore, the connection to manifold learning may explain why these generative models fail when they do. Known as *posterior collapse* in VAEs (Alemi et al., 2017; Zhao et al., 2017; He et al., 2019; Razavi et al., 2019) and *mode collapse* in GANs (Goodfellow, 2017), both describe cases where the forward/reverse mapping to/from Euclidean space collapses large parts of the input to a single output. This violates the bijective requirement of the homeomorphic mapping. It also results in degenerate latent spaces and poor generative performance. A major cause of such failings is when

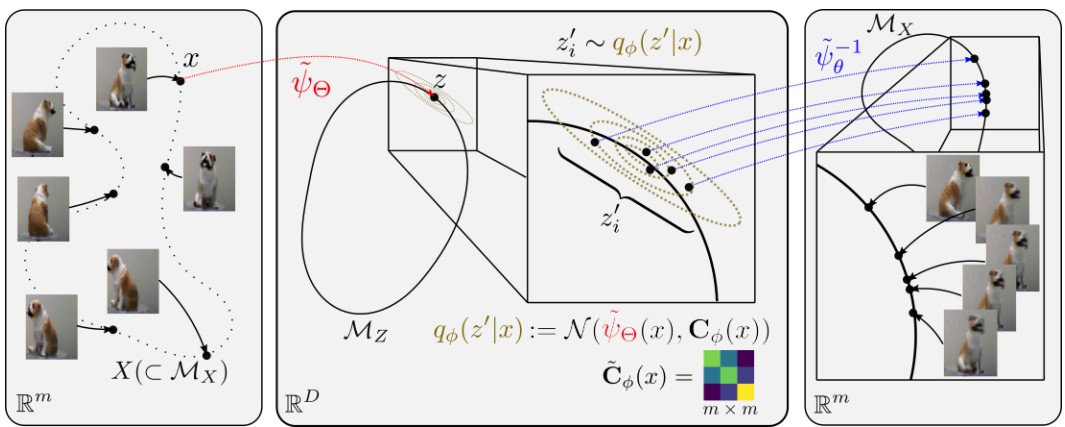

Figure 1: A diagram depicting one step of the diffusion process modeled by the variational diffusion autoencoder (VDAE). The diffusion and inverse diffusion maps $\psi, \psi^{-1}$, as well as the covariance $\mathbf{C}$ of the random walk on $\mathcal{M}_Z$, are all approximated by neural networks.

the geometries of the prior and target data do not agree. We explore this issue of *prior mismatch* and previous treatments of it in Section 3.

Given their connection to manifold learning, it is natural to look to classical approaches in the field for ways to improve VAEs. One of the most principled methods is spectral learning (Schölkopf et al., 1998; Roweis & Saul, 2000; Belkin & Niyogi, 2002) which involves describing data from a manifold $X \subset \mathcal{M}_X$ by the eigenfunctions of a kernel on $\mathcal{M}_X$. We focus specifically on DMs, where Coifman & Lafon (2006) show that normalizations of the kernel approximate a very specific diffusion process, the heat kernel over $\mathcal{M}_X$. A crucial property of the heat kernel is that, like its physical analogue, it defines a diffusion process that has a uniform stationary distribution — in other words, drawing from this stationary distribution draws uniformly from the data manifold. Moreover, Jones et al. (2008) established another crucial property of DMs, namely that distances in local neighborhoods in the eigenfunction space are nearly isometric to corresponding geodesic distances on the manifold. However, despite its strong theoretical guarantees, DMs are poorly equipped for large scale generative modeling as they are not easily scalable and do not provide an inverse mapping from the intrinsic feature space.

In this paper we address issues in variational inference and manifold learning by combining ideas from both. Theory in manifold learning allows us to better recognize *prior mismatch*, whereas variational inference provides a method to learn the difficult to approximate inverse diffusion map.

**Our contributions: 1**) We introduce the locally bi-Lipschitz property, a sufficient condition for a homeomorphism, for measuring the stability of a mapping between latent and data distributions. **2**) We introduce VDAEs, a class of variational autoencoders whose encoder-decoder feedforward pass approximates the diffusion process on the data manifold with respect to a user-defined kernel $k$. **3**) We show that deep neural networks are capable of learning such diffusion processes, and **4**) that networks approximating this process produce random walks that have certain desirable properties, including well defined transition and stationary distributions. **5**) Finally, we demonstrate the utility of the VDAE framework on a set of real and synthetic datasets, and show that they have superior performance and satisfy the locally bi-Lipschitz property where GANs and VAEs do not.

## 2 BACKGROUND

**Variational inference** (VI, Jordan et al. (1999); Wainwright et al. (2008)) is a machine learning method that combines Bayesian statistics and latent variable models to approximate some probability density $p(x)$. VI assumes and exploits a latent variable structure in the assumed data generation process, that the observations $x \sim p(x)$ are conditionally distributed given unobserved latent vari-

ables $z$. By modeling the conditional distribution, then marginalizing over $z$, as in

$$p_\theta(x) = \int_z p_\theta(x|z)p(z)dz, \tag{1}$$

we obtain the model evidence, or likelihood that $x$ could have instead been drawn from $p_\theta(x)$. Maximizing Eq. 1 leads to an algorithm for finding likely approximations of $p(x)$. As the cost of computing this integral scales exponentially with the dimension of $z$, we instead maximize the evidence lower bound (ELBO):

$$\log p_\theta(x) \geq -D_{KL}(q(z|x)||p(z)) + \mathbb{E}_{z \sim q(z|x)}[\log p_\theta(x|z)], \tag{2}$$

where $q(z|x)$ is usually an approximation of $p_\theta(z|x)$. Optimizing the ELBO is sped up by taking stochastic gradients (Hoffman et al., 2013), and further accelerated by learning a global function approximator $q_\phi$ in an autoencoding structure (Kingma & Welling, 2013).

**Diffusion maps** (DMs, Coifman & Lafon (2006)) on the other hand, are a class of kernel methods that perform non-linear dimensionality reduction on a set of observations $X \subseteq \mathcal{M}_X$, where $\mathcal{M}_X$ is the data manifold. Given a symmetric and positive kernel $k$, DM considers the induced random walk on the graph of $X$, where given $x, y \in X$, the transition probabilities $p(y|x) = p(x, y)$ are row normalized versions of $k(x, y)$. Moreover, the diffusion map $\psi$ embeds the data $X \in \mathbb{R}^m$ into the Euclidean space $\mathbb{R}^D$ so that the diffusion distance is approximated by Euclidean distance. This is a powerful property, as it allows the arbitrarily complex random walk induced by $k$ on $\mathcal{M}_X$ to become an isotropic Gaussian random walk on $\psi(\mathcal{M}_X)$.

**SpectralNet** is an algorithm introduced by algorithm in Shaham et al. (2018b) to speed up the diffusion map. Until recently, the method $\psi_k$ could only be computed via the eigendecomposition of $K$. As a result, DMs were only be tractable for small datasets, or on larger datasets by combining landmark-based estimates and Nystrom approximation techniques. However, Shaham et al. (2018b) propose approximations of the function $\psi$ itself in the case that the kernel $k$ is symmetric. In particular, we will leverage *SpectralNet* to enforce our diffusion embedding prior.

**Locally bi-lipschitz coordinates by kernel eigenfunctions**. (Jones et al. (2008)) analyzed the construction of local coordinates of Riemannian manifolds by Laplacian eigenfunctions and diffusion map coordinates. They establish, for all $x \in X$, the existence of some neighborhood $U(x)$ and $d$ spectral coordinates given $U(x)$ that define a bi-Lipschitz mapping from $U(x)$ to $\mathbb{R}^d$. With a smooth compact Riemannian manifold, $U(x)$ can be chosen to be a geodesic ball with radius a constant multiple of the *inradius* (the radius of the largest possible ball around $x$ without intersecting with the manifold boundary), where the constant is uniform for all $x$, but the indices of the $d$ spectral coordinates as well as the local bi-Lipschitz constants may depend on $x$. Specifically, the Lipschitz constants involve inverse of the inradius at $x$ multiplied again by some global constants. For completeness we give a simplified statement of the Jones et al. (2008) result in the supplementary material.

Using the compactness of the manifold, one can always cover the manifold with $m$ many neighborhoods (geodesic balls) on which the bi-Lipschitz property in Jones et al. (2008) holds. As a result, there are a total of $D$ spectral coordinates, $D \leq md$ (in practice $D$ is much smaller than $md$, since the selected spectral coordinates in the proof of Jones et al. (2008) tend to be low-frequency ones, and thus the selection on different neighborhoods tend to overlap), such that on each of the $m$ neighborhoods, there exists a subset of $d$ spectral coordinates out of the $D$ ones which are bi-Lipschitz on the neighborhood, and the Lipschitz constants can be bounded uniformly from below and above.

## 3 MOTIVATION AND RELATED WORK

Our proposed measure and model is motivated by degenerate latent spaces and poor generative performance in a variational inference framework arising from *prior mismatch*: when the topologies of the data and prior distributions do not agree. In real world data, this is usually due to two factors: first, when the dimensionalities of the distributions do not match, and second, when the geometries do not match. It is easy to see that homeomorphisms between the distributions will not exist in either case: pointwise correspondences cannot be established, thus the bijective condition cannot be met. As a result, the model has poor generative performance — for each point not captured in the pointwise correspondence, the latent or generated distribution loses expressivity.

Though the default choice of Gaussian distribution for $p(z)$ is mathematically elegant and computationally expedient, there are many datasets, real and synthetic, for which this distribution is ill-suited. It is well known that spherical distributions are superior for modeling directional data (Fisher et al., 1993; Mardia, 2014), which can be found in fields as diverse as bioinformatics (Hamelryck et al., 2006), geology (Peel et al., 2001), material science (Krieger Lassen et al., 1994), natural image processing (Bahlmann, 2006), and simply preprocessed datasets[1]. Additionally observe that no homeomorphism exists between $\mathbb{R}^k$ and $\mathcal{S}^1$ for any $k$. For data distributed on more complex manifolds, the literature is sparse due to the difficult nature of such study. However, the manifold hypothesis is well-known and studied (Narayanan & Mitter, 2010; Fefferman et al., 2016).

Previous research on alleviating *prior mismatch* exists. Davidson et al. (2018); Xu & Durrett (2018) consider VAEs with the von-Mises Fisher prior, a geometrically hyperspherical prior. (Rey et al., 2019) further model arbitrarily complex manifolds as priors, but require explicit knowledge of the manifold (i.e. its projection map, scalar curvature, and volume). Finally, Tomczak & Welling (2017) consider mixtures of any pre-existing priors. But while these methods increase the expressivity of the priors available, they do not prescribe a method for choosing the prior itself. That responsibility still lies with the user.

Convserly, our method chooses the best prior automatically. To our knowledge, ours is the first to take a data-driven approach to prior selection. By using some data to inform the prior, we not only guarantee the existence of a homeomorphism between data and prior distributions, we explicitly define it by the learned diffusion map $\tilde{\psi}$.

## 4 METHOD

In this section we propose VDAEs, a variational inference method that, given the data manifold $\mathcal{M}_X$, observations $X \subset \mathcal{M}_X$, and a kernel $k$, models the geometry of $X$ by approximating a random walk over the latent diffusion manifold $\mathcal{M}_Z := \psi(\mathcal{M}_X)$. The model is trained by maximizing the *local evidence*: the evidence (i.e. log-likelihood) of each point given its random walk neighborhood. Points are generated from the trained model by sampling from $\pi$, the stationary distribution of the resulting random walk.

Starting from some point $x \in X$, we can roughly describe one step of the walk as the composition of three functions: **1**) the approximate diffusion map $\tilde{\psi}_\Theta : \mathcal{M}_X \to \mathcal{M}_Z$, **2**) a sampling procedure from the learned diffusion process $z' \sim q_\phi(z'|x) = \mathcal{N}(\tilde{\psi}_\Theta(x), \tilde{\mathbf{C}}_\phi)$ on $\mathcal{M}_Z$, and **3**) the learned inverse diffusion map $\tilde{\psi}_\theta^{s-1} : \mathcal{M}_Z \to \mathcal{M}_X$ that produces $x' \sim p(x'|z') = \mathcal{N}(\tilde{\psi}_\theta^{-1}(z'), cI)$ where the constant $c$ is user-defined and fixed.

We rely crucially on three advantages of our latent space $\tilde{\psi}_\Theta(X)$: **a**) that it is well-defined (given the first $D$ eigenvalues of $k$ are distinct), **b**) well-approximated (given SpectralNet) and **c**) that Euclidean distances in $\mathcal{M}_Z$ approximate single-step random walk distances on $\mathcal{M}_X$ (see Section 2 and Coifman & Lafon (2006)). Thus the transition probabilities induced by $k$ can be approximated by Gaussian kernels [2] in $\mathcal{M}_Z$.

Therefore, to model a diffusion random walk over $\mathcal{M}_Z$, we must learn the functions $\tilde{\psi}_\Theta, \tilde{\psi}_\theta^{-1}, \tilde{\mathbf{C}}_\phi$ that approximate the diffusion map, its inverse, and the covariance of the random walk on $\mathcal{M}_Z$, at all points $z \in \mathcal{M}_Z$. SpectralNet gives us $\tilde{\psi}_\Theta$. To learn $\tilde{\psi}_\theta^{-1}$ and $\tilde{\mathbf{C}}_\phi$, we use variational inference.

### 4.1 THE LOWER BOUND

Formally, let us define $U_x := B_d(x, \delta) \cap \mathcal{M}_X$, where $B_d(x, \delta)$ as the $\delta$-ball around $x$ with respect to $d(\cdot, \cdot)$, the diffusion distance on $\mathcal{M}_Z$. For each $x \in X$, we define the *local evidence* of $x$ as

$$\mathbb{E}_{x' \sim p(x'|x)|_{U_x}} \log p_\theta(x'|x), \tag{3}$$

---

[1] Any dataset where the data points have been normalized to be unit length becomes a subset of a hypersphere.

[2] Importantly, note that the choice of a Gaussian kernel in the latent space is ***not*** dependent on the choice of $k$. We have this invariance due to the aforementioned property of diffusion embeddings.

where $p(x'|x)|_{U_x}$ is the restriction of $p(x'|x)$ to $U_x$. The resulting *local evidence lower bound* is:

$$\log p_\theta(x'|x) \geq \underbrace{-D_{KL}(q_\phi(z'|x)||p_\theta(z'|x))}_{\text{divergence of random walk distributions}} + \underbrace{\mathbb{E}_{z' \sim q_\phi(z'|x)} \log p_\theta(x'|z')}_{\text{neighborhood reconstruction error}}. \quad (4)$$

Note that the *neighborhood* reconstruction error should be differentiated from the *self* reconstruction error that is in VAEs. Eq. 4 produces the empirical loss function:

$$\tilde{\mathcal{L}}_{\text{DVAE}} = -D_{KL}(q_\phi(z'|x)||p_\theta(z'|x)) + \log p_\theta(x'|z_i'), \quad (5)$$

where $z_i' = g_{\phi,\Theta}(x, \epsilon_i)$, $\epsilon_i \sim \mathcal{N}(0, I)$. $g_{\phi,\Theta}$ is the deterministic, differentiable function, depending on $\tilde{\psi}_\Theta$ and $\tilde{\mathbf{C}}_\phi$, that generates $q_\phi$ by the reparameterization trick [3] (Kingma & Welling, 2013).

---

**Algorithm 1** VDAE training

---

$\Theta, \phi, \theta \leftarrow$ Initialize parameters
Obtain parameters $\Theta$ for the approximate diffusion map $\tilde{\psi}_\Theta$ by Shaham et al. (2018b)
**while** not converged **do**
    $A \leftarrow$ Random minibatch from $X$
    **for** $x \in A$ **do**
        $z' \sim p_\phi(z'|\tilde{\psi}_\Theta(x))$                 ▷ Take one step of the diffusion random walk
        $x' \leftarrow \arg\min_{y \in A \setminus \{x\}} |\tilde{\psi}_\Theta(y) - z'|_d^2$      ▷ Find approximate nearest neighbor(s)
        $g \leftarrow g + \frac{1}{|A|} \nabla_{\phi,\theta} \log p_\theta(x'|x)$      ▷ Compute gradients of the loss, i.e. Eq. equation 4
    Update $\phi, \theta$ using $g$

---

## 4.2 THE SAMPLING PROCEDURE

Here we discuss the algorithm for generating data points from $p(x)$. Composing $q_\phi(z'|x)(\approx p_\theta(z'|x))$ with $p_\theta(x'|z')$ gives us an approximation of $p_\theta(x'|x)$. Then the simple, parallelizable, and fast random walk based sampling procedure naturally arises: initialize with an arbitrary point on the manifold $x_0 \in \mathcal{M}_X$, then pick suitably large $N$ and for $n = 1, \ldots, N$ draw $x_n \sim p(x|x_{n-1})$. Eventually, our diffusion random walk converges on its stationary distribution $\pi$. By Coifman & Lafon (2006), this is guaranteed to be the uniform distribution on the data manifold. See Section 6.2 for examples of points drawn from this procedure.

## 4.3 A PRACTICAL IMPLEMENTATION

We now introduce a practical implementation VDAEs, considering the case where $\tilde{\psi}_\Theta(x)$, $q_\phi(z'|x)$ and $p_\theta(x'|z')$ are neural network functions, as they are in VAEs and SpectralNet, respectively.

The **neighborhood reconstruction error**. Since $q_\phi(z'|x)$ models the neighborhood of $\tilde{\psi}_\Theta(x)$, we may sample $q_\phi$ to obtain $z'$ (the neighbor of $x$ in the latent space). This gives $p_\theta(x'|x) \approx \tilde{\psi}_\theta^{-1}(q_\phi(z'|x))$, where $\psi^{-1}$ exists due to the bi-Lipschitz property. We can efficiently approximate $x' \in \mathcal{M}_X$ by considering the closest embedded data point $\tilde{\psi}_\Theta(x) \in \mathcal{M}_Z$ to $z' = \tilde{\psi}_\Theta(x')$. This is because Euclidean distance on $\mathcal{M}_Z$ approximates the diffusion distance on $\mathcal{M}_X$. In other words, $x' \sim p_\theta(x'|x) \approx \tilde{\psi}_\theta^{-1}(q_\phi(z'|x))$ which we approximate empirically by

$$x' \approx \arg\min_{y \in A} d(\tilde{\psi}_\Theta(y), z'), \quad z' \sim q_\phi(z'|x), \quad (6)$$

where $A \subseteq X$ is the training batch.

On the other hand, the **divergence of random walk distributions** $-D_{KL}(q_\phi(z'|x)||p_\theta(z'|x))$ can be modeled simply as the divergence of two Gaussian kernels defined on $\mathcal{M}_Z$. Though $p_\theta(z'|x)$ is intractable, the diffusion map $\psi$ gives us the diffusion embedding $Z$, which is an approximation of the true distribution of $p_\theta(z'|x)$ in a neighborhood around $z = \psi(x)$. We estimate the first and

---

[3]Though $q$ depends on $\phi$ and $\Theta$, we will use $q_\phi := q_{\phi,\Theta}$ to be consistent with existing VAE notation and to indicate that $\Theta$ is not learned by VI.

second moments of this distribution in $\mathbb{R}^D$ by computing the local Mahalanobis distance of points in the neighborhood. Then, by minimizing the KL divergence between $q_\phi(z'|x)$ and the one implied by this Mahalanobis distance, we obtain the loss:

$$-D_{KL}(q_\phi(z'|x)||p_\theta(z'|x)) = -\log\frac{|\alpha\Sigma_*|}{|\mathbf{C}_\phi|} + d - tr\{(\alpha\Sigma_*)^{-1}\mathbf{C}_\phi\}, \tag{7}$$

where $\mathbf{C}_\phi(x)$ is a neural network function, $\Sigma_*(x) = \text{Cov}(B_d(\psi(x),\delta) \cap Z)$ is the covariance of the points in a neighborhood of $z = \psi(x) \in Z$, and $\alpha$ is a scaling parameter. Note that $\mathbf{C}_\phi(x)$ does not have to be diagonal, and in fact is most likely not. Combining Eqs. 6 and 7 we obtain Algorithm 1.

Now we consider the sampling procedure. Since we use neural networks to approximate $q_\phi(z'|x)$ and $p_\theta(x'|z')$, the generation procedure is highly parallelizable. We empirically observe the random walk enjoys rapid mixing properties — it does not take many iterations of the random walk to sample from all of $\mathcal{M}_Z$ [4]. This leads to Algorithm 2.

---

**Algorithm 2** VDAE sampling

---

$X_0 \leftarrow$ Initialize with points $X_0 \subset X$
$t \leftarrow 0$
**while** $p(X_0) \not\approx \pi$ **do**
    **for** $x_t \in X_t$ **do**
        $z_{t+1} \sim p_\phi(z'|\tilde{\psi}_\Theta(x_t))$                $\triangleright$ Take one step of the diffusion random walk
        $x_{t+1} \sim p_\theta(x|z_{t+1})$                           $\triangleright$ Map back into input space
    $t \leftarrow t+1$

---

## 5 THEORY

We theoretically prove that the desired inverse map $\psi^{-1}$ from spectral coordinate codes back to the manifold can be approximated by a decoder network, where the network complexity is bounded by quantities related to the intrinsic geometry of the manifold. This section relies heavily on the known bi-Lipschitz property of DMs Jones et al. (2008), which we are approximating with the VDAE latent space without the need for regularization.

### 5.1 THEOREMS

The theory for the capacity of the encoder to map $\mathcal{M}$ to the diffusion map space $\psi(\mathcal{M})$ has already been considered in Shaham et al. (2018a) and Mishne et al. (2017). We instead focus on the decoder, which requires a different treatment. The following theorem is proved in Appendix A.3, based upon the result in Jones et al. (2008).

**Theorem 1.** *Let $\mathcal{M}_X \subset \mathbb{R}^m$ be a smooth $d$-dimensional manifold, $\psi(\mathcal{M}_X) \subset \mathbb{R}^D$ be the diffusion map for $D \geq d$ large enough to have a subset of coordinates that are locally bi-Lipschitz. Let $\mathbf{X} = [X_1, \ ..., \ X_m]$ be the set of all $m$ extrinsic coordinates of the manifold. Then there exists a sparsely-connected ReLU network $f_N$, with $4DC_{\mathcal{M}_X}$ nodes in the first layer, $8dmN$ nodes in the second layer, and $2mN$ nodes in the third layer, and $m$ nodes in the output layer, such that*

$$\|\mathbf{X}(\psi(x)) - f_N(\psi(x))\|_{L^2(\psi(\mathcal{M}_X))} \leq \sqrt{m}C_\psi/\sqrt{N} \tag{8}$$

*where the norm is interpreted as $\|F\|^2_{L^2(\psi(\mathcal{M}))} := \int \|F(\psi(x))\|^2_2 d\psi(x)$. Here $C_\psi$ depends on how sparsely $X(\psi(x))\big|_{U_i}$ can be represented in terms of the ReLU wavelet frame on each neighborhood $U_i$, and $C_{\mathcal{M}_X}$ on the curvature and dimension of the manifold $\mathcal{M}_X$.*

Theorem 1 is complementary to the theorem in Shaham et al. (2018a), which provides guarantees for the encoder, as Theorem 1 demonstrates a similar approximation theoretic argument for the decoder. The proof is built on two properties of ReLU neural networks: 1) their ability to split curved domains into small, almost Euclidean patches, 2) their ability to build differences of bump functions

---

[4]For all experiments in Section 6, the number of steps required to draw from $\pi$ is less than 10.

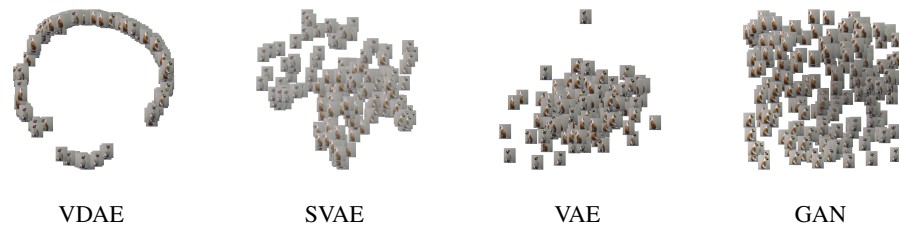

| VDAE | SVAE | VAE | GAN |

Figure 2: Reconstructed images from the rotating bulldog example plotted in the latent space of VDAE (left), Spherical VAE (SVAE, left-middle) and VAE (right-middle), and GAN (right)

on each patch, which allows one to borrow approximation results from the theory of wavelets on spaces of homogeneous type. The proof also crucially uses the bi-Lipschitz property of the diffusion embedding Jones et al. (2008). The key insight of Theorem 1 is that, because of the bi-Lipschitz property, the coordinates of the manifold in the ambient space $\mathbb{R}^m$ can be thought of as functions of the diffusion coordinates. We show that because each of coordinates function $X_i$ is a Lipschitz function, the ReLU wavelet coefficients of $X_i$ are necessarily $\ell^1$. This allows us to use the existing guarantees of Shaham et al. (2018a) to complete the desired bound.

We also discuss the connections between the distribution at each point in diffusion map space, $q_\phi(z|x)$, and the result of this distribution after being decoded through the decoder network $f_N(z)$ for $z \sim q_\phi(z|X)$. Similar to Singer & Coifman (2008), we characterize the covariance matrix $Cov(f_N(z)) := \mathbb{E}_{z \in q_\phi(z|x)}[f_N(z)f_N(z)^T]$. The following theorem is proved in Appendix A.3.

**Theorem 2.** *Let $f_N$ be a neural network approximation to $\mathbf{X}$ as in Theorem 1, such that it approximates the extrinsic manifold coordinates. Let $C \in \mathbb{R}^{m \times m}$ be the covariance matrix $C = \mathbb{E}_{z \in q_\phi(z|x)}[f_N(z)f_N(z)^T]$. Let $q_\phi(z|x) \sim N(\psi(x), \Sigma)$ with small enough $\Sigma$ that there exists a patch $U_{z_0} \subset \mathcal{M}$ around $z_0$ satisfying the bi-Lipschitz property of Jones et al. (2008), and such that $Pr(z \sim q_\phi(z|x) \notin \psi(U_{z_0})) < \epsilon$. Then the number of eigenvalues of $C$ greater than $\epsilon$ is at most $d$, and $C = J_{z_0}\Sigma J_{z_0}^T + O(\epsilon)$ where $J_{z_0}$ is the $m \times D$ Jacobian matrix at $z_0$.*

Theorem 2 establishes the relationship between the covariance matrices used in the sampling procedure and their image under the decoder $f_N$ to approximate $\psi^{-1}$. Similar to Singer & Coifman (2008), we are able to sample according to a multivariate normal distribution in the latent space. Thus, the resulting cloud in the data space is distorted (to first order) by the local Jacobian of the map $f_N$. The key insight of Theorem 2 is from combining this idea with the observation of Jones et al. (2008) that $\psi^{-1}$ depends locally on only $d$ of the coordinates in the $D$ dimensional latent space.

## 6 EXPERIMENTAL RESULTS

### 6.1 VIDEO OF ROTATING FIGURE

We consider the problem of generating new frames from a video of rigid movement. We take 200 frames of a color video (each frame is $100 \times 80 \times 3$) of a spinning bulldog Lederman & Talmon (2018). Due to the spinning of figure and the fixed background, this creates a low-dimensional approximately circular manifold.

We compare our method to VAE, the Wasserstein GAN Gulrajani et al. (2017) (with a bi-lipchitz constraint on the critic), and the hyperspherical VAE Davidson et al. (2018). For the VAE, we use a two dimensional Gaussian prior $p_\theta(z)$, such that $z \sim N(0, I_2)$. The noise injected to the GAN is drawn from a two dimensional uniform distribution $p_\theta(z)$, such that $z_i \sim U(0,1), i = 1, 2$. For the spherical VAE, we use a latent dimension of $D = 2$, which highlights the dimension mismatch issue that occurs with a spherical prior. This is a benefit of VDAE, even if we choose $D > d$ the latent embedding will still only be locally $d$ dimensional. We use the same architecture for all networks which consists of one hidden layer with 512 neurons, activation function for all networks are tanh. In Fig. 2, we present 300 generated samples, by displaying them on a scatter plot with coordinates corresponding to their latent dimensions $z_1$ and $z_2$.

### 6.2 DATA GENERATION FROM UNIFORMLY SAMPLED MANIFOLDS

In this series of experiments, we visualize the results of the sampling procedure in Algorithm 2 on three synthetic manifolds. As discussed in 4.2, we randomly select an initial seed point, then

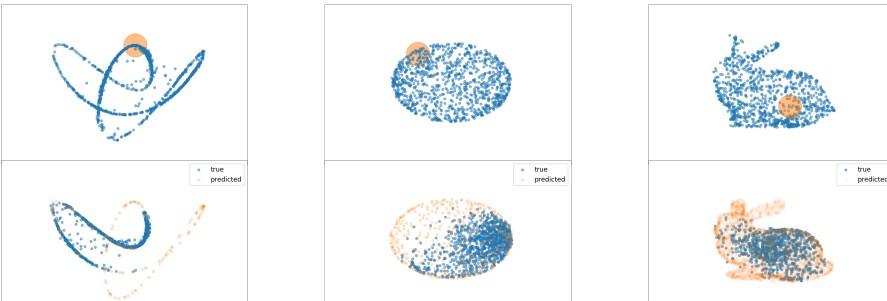

Figure 3: An example of distributions reconstructed from a random walk on $\mathcal{M}_Z$ (via Algorithm 2), given a single seed point drawn from $X$. (Bottom): An example of a single burst $p_\theta(x|z)$. The distributions are a loop (left), sphere (middle), and the Stanford bunny (right).

recursively sample from $p_\theta(x'|x)$ many times to simulate a random walk on the manifold. In the top row of Fig. 3, we highlight the location of the initial seed point, take 20 steps of the random walk, and display the resulting generated points on three learned manifolds. Clearly after a large number of resampling iterations, the algorithm continues to generate points on the manifold, and the distribution of sampled points converges to a uniform stationary distribution on the manifold. Moreover, this stationary distribution is reached very quickly. In the bottom row of the same Fig. 3, we show $p_\theta(x'|x)$ by sampling a large number of points sampled from the single seed point. As can be seen, a single step of $p_\theta(x'|x)$ covers a large part of the latent space. The architecture also uses one hidden layer of 512 neurons and tanh activations.

### 6.3 CLUSTER CONDITIONAL SAMPLING

In this section, we deal with the problem of generating samples from data with multiple clusters in an unsupervised fashion (i.e. no a priori knowledge of the cluster structure). Clustered data creates a problem for many generative models, as the topology of the latent space (i.e. normal distribution) differs from the topology of the data space with multiple clusters.

In our first experiment, we show that our method is capable of generating new points from a particular cluster given an input point from that cluster. This generation is done in an unsupervised fashion, which is a different setting from the approach of conditional VAEs Sohn et al. (2015) that require training labels. We demonstrate this property on MNIST in Figure 4, and show that the newly generated points after short diffusion time remain in the equivalent class to the seeded image. Here the architecture is a standard fully convolutional architecture. Details can be found in Appendix A.4.



Figure 4: An example of cluster conditional sampling with our method, given a seed point (top left of each image grid). The DVAE is able to produce examples via the random walk that stay approximately within the cluster of the seed point, without any supervised knowledge of the cluster.

The problem of addressing difference in topologies between the latent space of a generative model and the output data has been acknowledged in recent works about rejection sampling (Azadi et al., 2018; Turner et al., 2018). Rejection sampling of neural networks consists of generating a large collection of samples using a standard GAN, and then designing a probabilistic algorithm to decide in a *post-hoc* fashion whether the points were truly in the support of the data distribution $p(x)$.

In the following experiment, we compare to the standard example in the generative model literature. The data consists of nine bounded spherical densities with significant minimal separation, lying on a $5 \times 5$ grid. A standard GAN or VAE struggles to avoid generating points in the gaps between

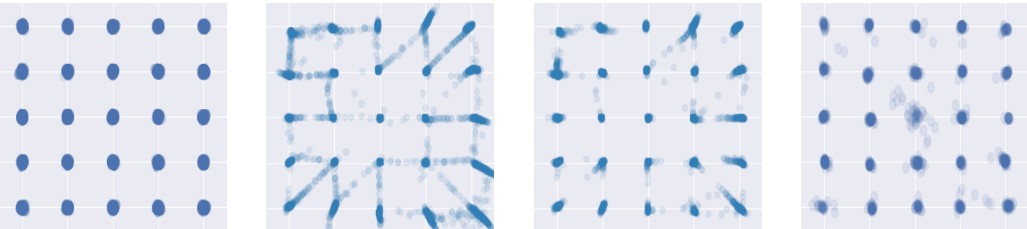

Figure 5: Comparison between GAN, DRS-GAN, and our samples on a $5 \times 5$ Gaussian grid. GAN and DRS-GAN samples taken from Azadi et al. (2018). Shown from left-right are Original, GAN, DRS-GAN, and our method.

these densities, and thus requires the post-sampling rejection analysis. On the other hand, our model creates a latent space that separates each of these clusters into their own features and only generates points that exist in the neighborhood of training data. Figure 5 clearly shows that this results in significantly fewer points generated in the gaps between clusters, as well as eliminating the need to generate additional points that are not in final generated set. Our VDAE architecture here uses one hidden layer of 512 neurons and tanh activations. GAN and DRS-GAN architectures are as described in Azadi et al. (2018).

Table 1: Mean and standard deviation of the local bi-Lipschitz constant, local to k-nearest neighborhoods on MNIST where $k = 3, 5, 10, 100, 1000$. Lower is better.

| Method | k=3 | 5 | 10 | 100 | 1000 |
|---|---|---|---|---|---|
| GAN | $2.24 \pm 0.35$ | $2.32 \pm 0.29$ | $2.5 \pm 0.34$ | $2.8 \pm 0.42$ | $3.11 \pm 0.44$ |
| VAE | $8.63 \pm 2.53$ | $8.93 \pm 2.55$ | $9.17 \pm 2.59$ | $9.51 \pm 2.64$ | $9.78 \pm 2.6$ |
| SVAE | $47.2 \pm 19.1$ | $50.3 \pm 19.0$ | $52.6 \pm 19.1$ | $55.2 \pm 19.5$ | $58.5 \pm 19.4$ |
| VDAE | $\mathbf{1.92 \pm 0.5}$ | $\mathbf{2.05 \pm 0.49}$ | $\mathbf{2.22 \pm 0.50}$ | $\mathbf{2.64 \pm 0.52}$ | $\mathbf{2.87 \pm 0.516}$ |

### 6.4 Empirical evaluation of the local bi-Lipschitz measure

Here we describe a practical method for computing the local bi-Lipschitz property, then use it to evaluate several methods on the MNIST dataset. Let $Z$ and $X$ be metric spaces and $f : Z \to X$. We define, for each $z \in Z$ and $k \in \mathbb{N}$, the function $\mathtt{bilip}_k(z)$:

$$\mathtt{bilip}_k(z) = \min K \text{ s.t. } \frac{1}{K} \leq \frac{d_X(f(z), f(z'))}{d_Z(z, z')} \leq K, \forall \, z' \in U_{z,k} \cap Z$$

where $Z := f^{-1}(X)$ is the latent embedding of our dataset $X$ [5], $d_X$ and $d_Z$ are metrics on $X$ and $Z$, and $U_{z,k}$ is the k-nearest neighborhood of $z$. Intuitively, increasing values of $K$ can be thought of as an increasing tendency of the learned map to *stretch* or *compress* regions of space. By analyzing various statistics of the local bi-Lipschitz measure evaluated at all points of a latent space $Z$, we can gain insight into how well-behaved a homeomorphism $f$ is. In Table 1 we report the mean and standard deviation, over 10 runs, of the local bi-Lipschitz property for several methods trained on the MNIST dataset.

The comparison is between the Wassertein GAN (WGAN), the VAE, the hyperspherical VAE (SVAE), and our method. We use standard architectures prescribed by their respective papers to train the methods. For our method we use a single 500 unit hidden layer network architecture with ReLU nonlinearities for both the encoder and decoder.

By constraining our latent space to be the diffusion embedding of the data, our method finds a mapping that automatically enjoys the homeomorphic properties of an ideal mapping, and this is reflected in the low values of the local bi-Lipschitz constant. Conversely, other methods do not consider the topology of the data in the prior distribution. This is especially appparent in the VAE

---

[5]For VAE, SVAE, and our method, these are the means of the posterior distributions. For GAN it is points drawn from the $\mathcal{N}(0, 1)$ prior.

and SVAE, which must generate from the entirety of the input distribution $X$ since they minimize a reconstruction loss. Interestingly, the mode collapse tendency of GANs alleviate the pathology of the bi-Lipschitz constant by allowing the GAN to focus on a subset of the distribution — but this comes at the cost, of course, of collapsing to a few modes of the dataset. Our method is able to reconstruct the entirety of $X$ while simultaneously maintaining a low local bi-Lipschitz constant.

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

# A   APPENDIX

## A.1   DERIVATION OF LOCAL EVIDENCE LOWER BOUND (EQ. 4)

We begin with taking the log of the random walk transition likelihood,

$$\log p_\theta(x'|x) = \log \int_z' p_\theta(x', z'|x)dz' \tag{A.1}$$

$$= \log \int_z p_\theta(x'|z', x)p(z'|x)\frac{q(z')}{q(z')}dz' \tag{A.2}$$

$$= \log \mathbb{E}_{z'\sim q(z')} \left[ p_\theta(x'|z', x)\frac{p(z'|x)}{q(z')} \right] \tag{A.3}$$

$$\geq \mathbb{E}_{z'\sim q(z')} \left[ \log p_\theta(x'|z', x) \right] + \mathbb{E}_{z'\sim q(z')} \left[ \log \frac{p(z'|x)}{q(z')} \right] \tag{A.4}$$

$$\geq \mathbb{E}_{z'\sim q(z')} \left[ \log p_\theta(x'|z', x) \right] + D_{KL}[q(z')||p(z'|x)] \tag{A.5}$$

where $q(z')$ is an arbitrary distribution. We let $q(z')$ to be the conditional distribution $q(z'|x)$. Furthermore, if we make the simplifying assumption that $p_\theta(x'|z', z) = p_\theta(x'|z')$, then we obtain Eq. 4

$$\log p_\theta(x'|x) \geq -D_{KL}(q_\phi(z'|x)||p_\theta(z'|x)) + \mathbb{E}_{z'\sim q_\phi(z'|x)} \log p_\theta(x'|z'). \tag{A.6}$$

## A.2   RESULTS IN JONES ET AL. (2008)

To state the result in Jones et al. (2008), we need the following set-up:

(C1) $\mathcal{M}$ is a $d$-dimensional smooth compact manifold, possibly having boundary, equipped with a smooth (at least $C^2$) Riemannian metric $g$;

We denote the geodesic distance by $d_\mathcal{M}$, and the geodesic ball centering at $x$ with radius $r$ by $B_\mathcal{M}(x, r)$. Under (C1), for each point $x \in \mathcal{M}$, there exists $r_\mathcal{M}(x)$ which is the inradius, that is, $r$ is the largest number s.t. $B_\mathcal{M}(x, r)$ is contained $\mathcal{M}$.

Let $\triangle_\mathcal{M}$ be the Laplacian-Beltrami operator on $\mathcal{M}$ with Neumann boundary condition, which is self-adjoint on $L^2(M, \mu)$, $\mu$ being the Riemannian volume given by $g$. Suppose that $\mathcal{M}$ is re-scaled to have volume 1. The next condition we need concerns the spectrum of the manifold Laplacian

(C2) $\triangle_\mathcal{M}$ has discrete spectrum, and the eigenvalues $\lambda_0 \leq \lambda_1 \leq \cdots$ satisfy the Weyl's estimate, i.e. exists constant $C$ which only depends on $\mathcal{M}$ s.t.

$$|\{j : \lambda_j \leq T\}| \leq CT^{d/2}.$$

Let $\psi_j$ be the eigenfunction associated with $\lambda_j$, $\{\psi_j\}_j$ form an orthonormal bases of $L^2(M, \mu)$. The last condition is

(C3) The heat kernel (defined by the heat equation on $\mathcal{M}$) has the spectral representation as

$$K_t(x, y) = \sum_{j=0}^\infty e^{-t\lambda_j}\psi_j(x)\psi_j(y).$$

**Theorem 3** (Theorem 2 Jones et al. (2008), simplified version). *Under the above setting and assume (C1)-(C2), then there are positive constants $c_1, c_2, c_3$ which only depend on $\mathcal{M}$ and $g$, s.t. for any $x \in \mathcal{M}$, $r_\mathcal{M}(x)$ being the inradius, there are $d$ eigenfunctions of $\triangle_\mathcal{M}$, $\psi_{j_1}, \cdots, \psi_{j_d}$, which collectively give a mapping $\Psi : \mathcal{M} \to \mathbb{R}^d$ by*

$$\Psi_x(x) = (\psi_{j_1}(x), \cdots, \psi_{j_d}(x))$$

*satisfying that $\forall y, y' \in B(x, c_1 r_\mathcal{M}(x))$,*

$$c_2 r_\mathcal{M}(z)^{-1}d_\mathcal{M}(y, y') \leq \|\Psi_x(y) - \Psi_x(y')\| \leq c_3 r_\mathcal{M}(z)^{-1-d/2}d_\mathcal{M}(y, y').$$

*That is, $\Psi$ is bi-Lipschitz on the neighborhood $B(x, c_1 r_\mathcal{M}(x))$ with the Lipschitz constants indicated as above. The subscript $x$ in $\Psi_x$ emphasizes that the indices $j_1, \cdots, j_d$*

*may depend on $x$.*

### A.3 Proofs

*Proof of Theorem 1.* The proof of Theorem 1 is actually a simple extension of the following theorem, Theorem 4, which needs to be proved for each individual extrinsic coordinate $X_k$, hence the additional factor of $m$ coming from the L2 norm of $m$ functions. □

**Theorem 4.** *Let $\mathcal{M} \subset \mathbb{R}^m$ be a smooth d-dimensional manifold, $\psi(\mathcal{M}) \subset \mathbb{R}^D$ be the diffusion map for $D \geq d$ large enough to have a subset of coordinates that are locally bi-Lipschitz. Let one of the $m$ extrinsic coordinates of the manifold be denoted $X(\psi(x))$ for $x \in \mathcal{M}$. Then there exists a sparsely-connected ReLU network $f_N$, with $4DC_\mathcal{M}$ nodes in the first layer, $8dN$ nodes in the second layer, and $2N$ nodes in the third layer, such that*

$$\|X - f_N\|_{L^2(\psi(\mathcal{M}))} \leq \frac{C_\psi}{\sqrt{N}} \tag{A.7}$$

*where $C_\psi$ depends on how sparsely $X(\psi(x))\big|_{U_i}$ can be represented in terms of the ReLU wavelet frame on each neighborhood $U_i$, and $C_\mathcal{M}$ on the curvature and dimension of the manifold $\mathcal{M}$.*

*Proof of Theorem 4.* The proof borrows from the main theorem of Shaham et al. (2018a). We adopt this notation and summarize the changes in the proof here. For a full description of the theory and guarantees for neural networks on manifolds, see Shaham et al. (2018a). Let $C_\mathcal{M}$ be the number of neighborhoods $U_i = B(x_i, \delta) \cap \mathcal{M}$ needed to cover $\mathcal{M}$ such that $\forall x, y \in U_i$, $(1 - \epsilon)\|x - y\| \leq d_\mathcal{M}(x, y) \leq (1 + \epsilon)\|x - y\|$. Here, we choose $\delta = \min(\delta_\mathcal{M}, \kappa^{-1}\rho)$ where $\delta_\mathcal{M}$ is the largest $\delta$ that preserves locally Euclidean neighborhoods and $\kappa^{-1}\rho$ is the smallest value from Jones et al. (2008) such that every neighborhood $U_i$ has a bi-Lipschitz set of diffusion coordinates.

Because of the locally bi-Lipschitz guarantee from Jones et al. (2008), we know for each $U_i$ there exists an equivalent neighborhood $\widetilde{\psi}(U_i)$ in the diffusion map space, where $\widetilde{\psi}(x) = [\psi_{i_1}(x), \quad ..., \quad \psi_{i_d}(x)]$. Note that the choice of these $d$ coordinates depends on the neighborhood $U_i$. Moreover, we know the Euclidean distance on $\psi(U_i)$ is locally bi-Lipschitz w.r.t. $d_\mathcal{M}(\cdot, \cdot)$ on $U_i$.

First, we note that as in Shaham et al. (2018a), the first layer of a neural network is capable of using $4D$ units to select the subset of $d$ coordinates $\widetilde{\psi}(x)$ from $\psi(x)$ for $x \in U_i$ and zeroing out the other $D - d$ coordinates with ReLU bump functions. Then we can define $X(\widetilde{\psi}(x)) = X(\psi(x))$ on $x \in U_i$.

Now to apply the theorem from Shaham et al. (2018a), we must establish that $X\big|_{U_i} : \widetilde{\psi}(U_i) \to \mathbb{R}$ can be written efficiently in terms of ReLU functions. Because of the manifold and diffusion metrics being bi-Lipschitz, we know at a minimum that $\widetilde{\psi}$ is invertible on $\widetilde{\psi}(U_i)$. Because of this invertibility, we will slightly abuse notation and refer to $X(\psi(x)) = X(x)$, where this is understood to be the extrinsic coordinate of the manifold at the point $x$ that cooresponds to $\psi(x)$. we also know that $\forall x, y \in U_i$,

$$
\begin{aligned}
|X(\widetilde{\psi}(x)) - X(\widetilde{\psi}(y))| &= |X(x) - X(y)| \\
&\leq \max_{z \in U_i} \|\nabla X(z)\| d(x, y) \\
&\leq \frac{\max_{z \in U_i} \|\nabla X(z)\|}{1 - \epsilon} \|\widetilde{\psi}(x) - \widetilde{\psi}(y)\|,
\end{aligned}
$$

where $\nabla X(z)$ is understood to be the gradient of $X(z)$ at the point $z \in \mathcal{M}$. This means $X(\widetilde{\psi}(x))$ is a Lipschitz function w.r.t. $\widetilde{\psi}(x)$. Because $X(\widetilde{\psi}(x))$ Lipschitz continuous, it can be approximated by step functions on a ball of radius $2^{-\ell}$ to an error that is at most $\frac{\max_{z \in U_i} \|\nabla X(z)\|}{1 - \epsilon} 2^{-\ell}$. This means the maximum ReLU wavelet coefficient is less than $\frac{\max_{z \in U_i} \|\nabla X(z)\|}{1 - \epsilon}(2^{-\ell} + 2^{-\ell+1})$. This fact, along with the fact that $\widetilde{\psi}(U_i)$ is compact, gives the fact that on $\widetilde{\psi}(U_i)$, set of ReLU wavelet coefficients is in $\ell^1$. And from Shaham et al. (2018a), if on a local patch the function is expressible in terms of ReLU wavelet coefficients in $\ell^1$, then there is an approximation rate of $\frac{1}{\sqrt{N}}$ for $N$ ReLU wavelet terms. □

*Proof of Theorem 2.* We borrow from Singer & Coifman (2008) to prove the following result. Given that the bulk of the distribution $q$ lies inside $\psi(U_{z_0})$, we can consider only the action of $f_N$ on $\psi(U_{z_0})$ rather than on the whole space. Because the geodesic on $U$ is bi-Lipschitz w.r.t. the Euclidean distance on the diffusion coordinates (the metric on the input space), we can use the results from Singer & Coifman (2008) and say that on $\psi(U_{z_0})$ the output covariance matrix is characterized by the Jacobian of the function $f_N$ mapping from Euclidean space (on the diffusion coordinates) to the output space, at the point $z_0$. So the covariance of the data lying insize $\psi(U_{z_0})$ is $J_{z_0} \Sigma J_{z_0}^T$, with an $O(\epsilon)$ perturbation for the fact that $\epsilon$ fraction of the data lies outside $\psi(U_{z_0})$.

The effective rank of $C$ being at most $d$ comes from the locally bi-Lipschitz property. We know $X(\psi(x))$ only depends on the $d$ coordinates $\widetilde{\psi}(x)$ as in the proof of Theorem 1, which implies $f_N(\psi(x))$ satisfies a similarly property if $f_N$ fully learned $X(\psi(x))$. Thus, while $J \in \mathbb{R}^{m \times D}$, it is at most rank $d$, which means $J \Sigma J^T$ is at most rank $d$ as well.

$\square$

## A.4 EXPERIMENT ARCHITECTURES

### A.4.1 CLUSTER CONDITIONAL SAMPLING WITH MNIST

We use the following encoder architecture:

- Conv2D(channels=64, strides=(1,1), kernel=4)
- Conv2D(channels=64, strides=(1,1), kernel=4)
- MaxPooling2D(pool_size=2)
- Conv2D(channels=64, strides=(1,1), kernel=4)
- Conv2D(channels=64, strides=(1,1), kernel=4)
- MaxPooling2D(pool_size=2)
- Dense(512, 'relu')
- Dense(10, 'linear')

and the following decoder architecture: We use the following decoder architecture:

- Dense(7 * 7, 'relu')
- Conv2DTranspose(channels=64, strides=(1,1), kernel=4)
- Conv2DTranspose(channels=64, strides=(1,1), kernel=4)
- UpSampling2D(pool_size=2)
- Conv2DTranspose(channels=64, strides=(1,1), kernel=4)
- Conv2DTranspose(channels=64, strides=(1,1), kernel=4)
- UpSampling2D(pool_size=2)

