# OpenReview forum: "Variational Diffusion Autoencoders with Random Walk Sampling"
_ICLR.cc/2020/Conference — Reject_

### Official Review · AnonReviewer3 · 2019-10-24
**Official Blind Review #3**

**Rating:** 8

**Review:**

** Summary
The paper studies the problem of density estimation and learning accurate generative models. The authors start from the observation that this problem has been approaches either using variational inference models, that scale very well but whose approximations may lead to degenerate results in practice, and diffusion maps, that scale poorly but are very effective in capturing the underlying data manifold. From here, the authors propose integrating the notion of random walk from diffusion maps into VAEs to avoid degenerate conditions. The proposed method is first defined in its generality, a practical implementation is presented, theoretical guarantees are provided, and empirical evidence of its effectiveness is reported.

** Evaluation
The starting point of the paper seems very solid: diffusion maps are capturing the geometry of data very effectively and bringing some of those characteristics into the more scalable approach of VAE is an interesting approach. In the proposed method, this translates into introducing a learned diffusion map from manifold to an Euclidean space into the inference part. As a result, the lower bound optimized by the method now contains local information about the accuracy of the one-step random walk. How this can be translated into a practical implementation is also convincing. My main concern is that the overall method is now approximating many different elements in the original formulation, such as the diffusion map, its inverse, and the covariance of the random walk. Although theory seems to support that as these approximation become more accurate the overall result is reliable, in practice I wonder how they could combine and deteriorate the final result.

The empirical validation is relatively simple but in my opinion it provides enough insights about the advantages of the proposed method compared to VAEs and GANs. More solid and extensive evaluation is definitely needed in the future to have a more thorough comparison and a more careful assessment of the limits of the proposed method, but at this stage, I think the evaluation is sufficient.

**Experience Assessment:**

I have read many papers in this area.

**Review Assessment: Checking Correctness Of Derivations And Theory:**

I did not assess the derivations or theory.

**Review Assessment: Checking Correctness Of Experiments:**

I assessed the sensibility of the experiments.

**Review Assessment: Thoroughness In Paper Reading:**

I read the paper at least twice and used my best judgement in assessing the paper.

---

> ### Author Response · Authors · 2019-11-09
> **Response to Blind Review #3**
>
> Thank you for this thoughtful review.  We can briefly address your main concern that several of the elements are approximations to the original formulation.  We agree that the SpectralNet embedding is an approximation of the kernel eigenvectors that form a diffusion map.  However, given that we still minimize the Rayleigh quotient to the same degree as the eigenvectors, then the latent space created by SpectralNet will be the same as the subspace spanned by the eigenvectors.  This would allow a carryover of similar guarantees from diffusion maps.  The closest guarantee that can be attained for SpectralNet, or truly for any network trained on manifold data, is that a small network can learn such a subspace with dependence only on the intrinsic dimension of the manifold (Shaham, Cloninger, Coifman, 2017).  So this justifies that the approximation should be close to the true original formulation.  This is the same argument for the inverse (VAE decoder) and the results in our Theorem 1.

---

### Official Review · AnonReviewer1 · 2019-10-24
**Official Blind Review #1**

**Rating:** 3

**Review:**

The paper proposes a new generative model for unsupervised learning, based on a diffusion random walk principle inspired by the manifold learning literature. The basic idea is to (probabilistically) map points to a latent space, perform a random walk in that space, and then map back to the original space again. Learning of the suitable maps is achieved by casting the problem in a variational inference framework.

The paper is generally well-written, and clearly states out its goals and motivation. Sections 1 - 3 in particular give a nice overview of the broader context of the paper, and its aim of borrowing ideas from manifold learning and variational autoencoders. The particular aim on using concepts from manifold learning to avoid mode collapse - corresponding to the underlying homeomorphism losing its bijectivity - is in particular intriguing.

The method itself is intuitive at a high level, although I did have some difficulty with Section 4:
- in 4.1, one begins by considering the local evidence. This requires drawing a point from U_x, which is defined to a be set. I presume this means one draws uniformly from this set?
- Eqn 3 does not apparently have the same structure as Eqn 2. In particular, the first term in (2) is a function of x, but for (3) it is not a function of x'. How does conditioning affect the ELBO?
- I was not sure how to interpret the statement that pθ(x'|x) ≈ ψ^{-1}(q(z'|x)). Do you mean the distribution is strongly concentrated around this value? Note also an extra "]" in the latter.
- Eqn 5 should presumably be an equality? Also, it was not clear what the d in |.|_d^2 means, and why one does not use ||.||^2.
- At a higher level, given that x' ~ U_x originally, why do we now draw x' ~ p(x'|x)?
- Arriving at 4.2, it was not clear what "The sampling procedure" refers to, i.e., which of the steps in 4.1 it is seeking to specify or augment. It would be useful to clearly lay out the objective function that is being optimised, and how this section fits into that.
- In 4.3, it seemed as if the discussion of the neighbourhood reconstruction error would be better placed in 4.1 itself. It appears to be a justification of the already-derived Eqn 5.
- Algorithm 2, is there a need to introduce Z_t? It is a bit confusing that, e.g., Z_1 is first written to in iteration 1 by g(Z_0, ε), and then by ψ(X_1) in the second iteration.

The authors also claim a contribution to be the identification of a principled measure to identify mismatch between latent and data distributions. This "bi-Lipschitz" property is only introduced in Sec 5.1, and the discussion is not too approachable to someone unfamiliar with the area. In particular:
- it is not clear how precisely the discussion in this section relates to the VDAE algorithm described in the previous section.
- precisely what quantity we are to compute so as to verify this condition remains elusive. The abstract and introduction made me expect that the property is practically verifiable, but it was not clear from this section whether it is so.
- the conclusion or key takeaway of this subsection was unclear. I gather that Jones et al. established the existence of a neighbourhood wherein one can define a bi-Lipschitz mapping to R^d for suitable d. But how does this relate to latent and data space mismatch?

The experiments show that the proposed method can generate meaningful samples for synthetic manifold data, as well as on the MNIST dataset. I would've preferred more discussion of the results in Sec 4.1. I also was hoping for a clearer illustration of mode-collapse problems on standard benchmarks for GANs, with comparison of results to, e.g., those of Wasserstein-GANs (beyond those in Sec 6.1) or other proposals that aim to mitigate mode collapse.

Minor comments:
- Fig 1, the text in the middle panel is hard to read in black and white.
- SpectralNet is mentioned a few times but never formally introduced.
- notationally, Section 4 is a little heavy. I would suggest considering to omit the subscript θ's in the function ψ and its inverse.
- when mentioning the "reparametrisation trick", please provide a citation.
- what is \mathbb{X} in Theorem 2?
- "completementary" -> "complementary"
- " rejection sampling Azadi et al. (2018); Turner et al. (2018)." -> " rejection sampling (Azadi et al. 2018; Turner et al. 2018)."
- some form of Conclusion would be appropriate.

**Experience Assessment:**

I do not know much about this area.

**Review Assessment: Checking Correctness Of Derivations And Theory:**

I assessed the sensibility of the derivations and theory.

**Review Assessment: Checking Correctness Of Experiments:**

I assessed the sensibility of the experiments.

**Review Assessment: Thoroughness In Paper Reading:**

I read the paper at least twice and used my best judgement in assessing the paper.

---

> ### Author Response · Authors · 2019-11-09
> **Response to Blind Review #1**
>
> Thank you for the thoughtful review. We will first address your questions and concerns about the bi-Lipschitz property.  You are correct that it is first brought up in Section 5, and that can be adjusted by moving it to a related works in Section 2 when discussing the diffusion map.  To clarify the connection, the bi-Lipschitz property is a property of diffusion maps and Laplacian Embeddings, as proved by Jones et al. Because of this, as long as we model the eigenfunctions of the kernel with SpectralNet, we don't have to regularize to maintain this property (and thus a stable homeomorphism). It comes up in section 5 so that we can prove there exists a stable inverse to map from the latent space back to the original data.
>
> As for verifying the condition, we will add to our manuscript a description of the exact condition.  The quantity to measure is $\|\psi(x) - \psi(y)\| / \|x - y\|$ for any points x,y such that $x,y \in B(x, r)$ for some radius $r$.  We can add an experimental verification of this measure in the appendix for the experiments run.  The closer this statistic is to 1, the closer you are to creating an isometry in the latent space.
>
> Also, we can address your questions about the sampling procedure.  We are sampling $x' \sim p(x' | x)$ when we input $x$ into the random walk VAE.  This $x'$ will be in the neighborhood of $x$.  The algorithm for sampling procedure is to draw a collection of $x'$ from the seed $x$, and then use these $x'$ as seeds to draw new points $x''$.  After several iterations of this procedure, we will have sampled points everywhere on the data manifold.  This is demonstrated in Figure 3, where the bottom line is one iteration of the sampling procedure, and the top line is the collection of points after enough iterations to converge to the distribution $p(x)$ on the manifold.
>
> We further address your concern about how we removed the dependence of the first term in (3) on x'. It is difficult to make theoretical guarantees about the tightness of the ELBO --- the lack of an approximation guarantee is a crucial property given up by variational inference methods (compared to MCMC techniques) in the interest of computational efficiency. However, we can make an intuitive argument. Note that $z' = \psi(x')$. Therefore, assuming we approximate the diffusion map to a reasonable accuracy, we do not lose much information when the dependence on x' is dropped.
>
> Finally, you noted that we should compare our method to more recent iterations of GANs that attempt to treat mode collapse. Perhaps it can be made more clear, but we do in fact do this: note that we use the Wasserstein GAN rather than the original GAN in our experiments.
>
> Thank you again for your comments. We are making the proposed changes to the manuscript, and will update you when they are available early next week.

---

### Official Review · AnonReviewer2 · 2019-11-02
**Official Blind Review #2**

**Rating:** 1

**Review:**

Summary: In this paper, a diffusion-based VAE is proposed. The authors introduced a non-linear dimension reduction method, diffusion map to standard VAE to encode neighborhood manifold information into the latent space before encoding. They proposed a lower-bound objective, similar to original VAE for content generation.

Pros: 1) A new VAE method is proposed to incorporate local manifold information into the latent space. 2) A sufficient condition to measure the consensus between latent and input data distributions. 3) Three empirical studies on the visualization of generated images.

Cons:
1) The writing could be significantly improved. There are a bunch of grammar errors and confusing notations. For example, “with many default priors the posterior/likelihood pair q(z|x)/p(x|z) can be viewed as an approximate homeomorphism” ->  “with many default priors,  the posterior/likelihood pair q(z|x)/p(x|z) that can be viewed as an approximate homeomorphism”. “In this paper address issues in variational inference and manifold learning” -> “In this paper, we address issues in variational inference ….”. “feedfoward pass” -> “feedforward” …. In algorithm 1, “X is a random batch from X”. Conditional probability are mixed with joint probabilities, “p(y|x) = p(x,y)”. I suggest the authors do careful proofreading.

2) The novelty in this paper is limited.  The diffusion VAE was proposed in Rey’19 https://arxiv.org/abs/1901.08991 with a similar random walk procedure with transition kernels on the manifold of input. However, the authors neither did any comparison with it nor provided convincing advantages over them.

3) The authors claimed the standard VAE has too many assumptions on the priors, likelihood, and posteriors. However, their framework also assumed Gaussian distribution on posteriors and likelihood, only eliminating the prior distribution, but at the expense of introducing an assumed kernel and eigendecomposition approximation.

3) The experiments are very limited, containing only 3 visualization results of 3 image generation tasks. The Fig 2 is difficult to read and interpret. How about the log-likelihood estimates from your approach compared with others?

**Experience Assessment:**

I have read many papers in this area.

**Review Assessment: Checking Correctness Of Derivations And Theory:**

I assessed the sensibility of the derivations and theory.

**Review Assessment: Checking Correctness Of Experiments:**

I assessed the sensibility of the experiments.

**Review Assessment: Thoroughness In Paper Reading:**

I read the paper at least twice and used my best judgement in assessing the paper.

---

> ### Author Response · Authors · 2019-11-09
> **Response to Blind Review #2**
>
> Thank you for your thoughtful review. We will respond below to the points you raised below.
> 1) We will definitely proofread the text thoroughly before the next revision. We appreciate your thorough reading and proposed edits.
> 2) We acknowledge that our method bears resemblance to that proposed in Rey ‘19. Both start from the same idea of sampling from manifold-supported latent spaces. However, the similarities end there. Crucial differences in the implementation of this idea result in drastically different algorithms. We mention this work briefly in Section 3. The key difference is that Rey ‘19 requires explicit knowledge of the manifold, including the projection map, the scalar curvature, and the volume of the manifold. These must be exactly specified beforehand. Conversely, our method only requires the dataset and a suitable kernel to capture pairwise similarities in the data. Moreover, the latent prior is user-defined in Rey ‘19. In other words, the user must have explicit knowledge of the topology of the data. Conversely, ours is learned automatically from the data by the diffusion map. For this reason we are able to intrinsically prevent prior mismatch.
> 3) Our posterior distribution is indeed Gaussian, but in the diffusion embedding space. Note that Euclidean distances in the diffusion embedding space approximate diffusion distances over the data manifold. Please see the original Diffusion Maps paper for more details, but in short, the diffusion distance is a measure of the similarity between two points based on the behavior of a diffusion process starting from either point. Therefore, our Gaussian distributed posterior defines a random walk that intrinsically respects this measure. This is a crucial property of our method that is carried over by its close ties to manifold learning.
> 4) As stated in R1’s review, we plan to add an experimental verification of the bi-Lipschitz measure in the appendix for the experiments run. The closer this statistic is to 1, the closer we are to creating an isometry in the latent space. In general, it is currently difficult to objectively evaluate the quality of generative models. The Inception Score and Frechet Inception Distance (FID) both work only on ImageNet models. Moreover, note that we optimize a very different likelihood: the neighborhood likelihood p(x’|x). (We are learning the random walk neighborhood, rather than the entire generative model.) Thus there is no direct comparison to existing VAE models.
>
> We are making the proposed changes to the manuscript, and will update you when they are available early next week.

---

### Author Response · Authors · 2019-11-15
**Updated manuscript**

Thank you again for your insightful reviews. We have made a few changes to the manuscript. Namely, four items:

#1: We made changes to manuscript to reflect many of the items suggested by R1. We also proofread the manuscript and fixed several grammatical errors pointed out by R2. Thank you both for your thorough reading!
#2: We added an extra experimental section (6.4) that compared the local bi-Lipschitz constant between our method, WGAN-GP, VAE, and SVAE. As the local bi-Lipschitz constant is a sufficient condition for a homeomorphism, we are able to directly evaluate how well-behaved the mappings of each method are. We show that our method has state-of-the-art performance by this new measure.
#3: We re-arranged the document to more clearly describe the locally bi-Lipschitz property. We moved 5.1 (the property) to Section 2 in background, and added some sentences in the intro and in Section 5 to clarify our measure.

Edit: Added some more changes

---

### Decision · Program_Chairs · 2019-12-19

**Decision:**

Reject

**Comment:**

This paper proposes to train latent-variable models (VAEs) based on diffusion maps on the data-manifold. While this is an interesting idea, there are substantial problems with the current draft regarding clarity, novelty and scalability. In its current form, it is unlikely that the proposed model will have  a substantial impact on the community.